# Main Predictors of COVID-19 Vaccination Uptake among Italian Healthcare Workers in Relation to Variable Degrees of Hesitancy: Result from a Cross-Sectional Online Survey

**DOI:** 10.3390/tropicalmed7120419

**Published:** 2022-12-06

**Authors:** Marco Trabucco Aurilio, Francesco Saverio Mennini, Cristiana Ferrari, Giuseppina Somma, Luca Di Giampaolo, Matteo Bolcato, Fabio De-Giorgio, Roberto Muscatello, Andrea Magrini, Luca Coppeta

**Affiliations:** 1Department of Medicine and Health Sciences “V.Tiberio”, University of Molise, 86100 Campobasso, Italy; 2Office of Medical Forensic Coordination, Italian National Social Security Institute (INPS), 00144 Rome, Italy; 3Economic Evaluation and HTA (EEHTA CEIS), Department of Economics and Finance, Faculty of Economics, University of Rome “Tor Vergata”, Via Columbia 2, 00133 Rome, Italy; 4Department of Occupational Medicine, University of Rome “Tor Vergata”, 00133 Rome, Italy; 5Department of Medicine and Science of Ageing, Specialization School of Allergy and Clinical Immunology, G. D’Annunzio University Chieti-Pescara, 66100 Chieti, Italy; 6Department of Neuroscience, University of Padua, 35121 Padua, Italy; 7Department of Healthcare Surveillance and Bioethics, Section of Legal Medicine, Università Cattolica del Sacro Cuore, 00168 Rome, Italy; 8Fondazione Policlinico Universitario IRCCS A. Gemelli, 00168 Rome, Italy

**Keywords:** vaccine hesitancy, COVID-19 vaccine, healthcare workers, vaccine confidence

## Abstract

Background: Hesitancy remains one of the major hurdles to vaccination, regardless of the fact that vaccines are indisputable preventive measures against many infectious diseases. Nevertheless, vaccine hesitancy or refusal is a growing phenomenon in the general population as well as among healthcare workers (HCWs). Many different factors can contribute to hesitancy to COVID-19 vaccination in the HCWs population, including socio-demographic characteristics (female gender, low socio-economical status, lower age), individual beliefs regarding vaccine efficacy and safety, as well as other factors (occupation, knowledge about COVID-19, etc.). Understanding the determinants of accepting or refusing the COVID-19 vaccination is crucial to plan specific interventions in order to increase the rate of vaccine coverage among health care workers. Methods: We conducted a cross-sectional online survey on HCWs in seventeen Italian regions, between 30 June and 4 July 2021, in order to collect information about potential factors related to vaccine acceptance and hesitancy. Results: We found an overall vaccine uptake rate of 96.4% in our sample. Acceptance was significantly related to job task, with physicians showing the highest rate of uptake compared to other occupations. At univariate analysis, the HCWs population’s vaccine hesitancy was significantly positively associated with fear of vaccination side effects (*p* < 0.01), and negatively related to confidence in the safety and efficacy of the vaccine (*p* < 0.01). Through multivariate analysis, we found that only the fear of possible vaccination side effects (OR: 4.631, *p* < 0.01) and the confidence in vaccine safety and effectiveness (OR: 0.35 *p* < 0.05) remained significantly associated with hesitancy. Conclusion: Action to improve operator confidence in the efficacy and safety of the vaccine should improve the acceptance rate among operators.

## 1. Introduction

The pandemic of Severe Acute Respiratory Syndrome Coronavirus 2 (SARS-CoV-2) first identified on 1st December 2019 in Wuhan, has since spread globally due to rapid virus diffusion through air droplets. Since December 2020, the World Health Organization (WHO) has approved the use of anti SARS-CoV-2 vaccines [1]. Vaccine hesitancy is a widespread phenomenon in the general population, despite the fact that vaccines are universally recognized as the most effective measure to prevent contagion, hospitalization, and death from infectious diseases. In 2019, the WHO declared it to be one of the worst ten threats to global health [2]. The term “vaccine hesitancy” is used when an individual is not sure of getting vaccinated. Vaccine hesitancy is defined as “apathy towards, deferral of or outright rejection of vaccines regardless of the availability and accessibility of the vaccination services” [3]. On the other hand, when an individual objects to getting vaccinated, the correct definition is “vaccine resistance” [4]. The major predictors for COVID-19 vaccine hesitancy are age, socioeconomic status, education, and health literacy [5]. Healthcare workers (HCWs) are generally recognized as the most trusted source of information regarding vaccines, but during recent decades, a growing trend toward vaccine refusal or delay has been reported among these operators globally.

Hesitancy is a complex phenomenon, and factors commonly related to vaccine hesitancy among the general population, such as confidence in vaccine safety and effectiveness, risk perception (complacency), and facilities for or barriers to accessing vaccination (convenience) can play a key role in vaccine acceptance in HCWs and in the general population [5,6]. In the actual COVID-19 epidemic, the achievement of a high rate of vaccinated HCWs has been proven to be an essential factor for limiting the impact of the pandemic waves on the health systems. Unvaccinated operators can put themselves and the most fragile patients at risk of nosocomial contagion. Common treatment options for COVID-19 include symptomatic over-the-counter medications, antiviral treatments (targeting specific parts of the virus to stop it from multiplying in the body and helping to prevent severe illness and death), and monoclonal antibodies (helping the immune system recognize and respond more effectively to the virus) [7]. Even if recent studies on COVID-19-infected subjects reported that the viral loads could be similar in vaccinated and unvaccinated subjects, the risk of infection is lower for vaccinated individuals and the decline in viral load is more rapid, resulting in a substantially reduced risk of onward transmission [8,9,10,11].

Specific factors associated with COVID-19 vaccine refusal among HCWs have not yet become fully understood. Recent reviews reported vaccine intention rates for COVID-19 among healthcare professionals ranging from 27.7% to 94.3% across different countries and areas, in relation to geographical, political, socio-demographic, and cultural factors [12,13].

To overcome the operators’ hesitancy, in many countries, mandatory SARS CoV-2 vaccination has been considered for HCWs. In Italy, by Law Decree No. 44 (1 April 2021) the government introduced the COVID-19 vaccination obligation for health professionals. Since a compulsory vaccination can improve vaccine uptake, but may undermine trust in the vaccination program and healthcare organization [14], the effect of these measures must be evaluated in follow-up studies [15].

Most of the largest studies conducted so far, including those mentioned above, were focused on the vaccine attitudes among HCWs; moreover, the rapidly evolving nature of vaccination programs decreases the ability to predict the real rate of vaccine uptake by HCWs.

In a recent Italian meta-analysis, it was found that the rate of hesitant operators was 13.1%, and that this number halved during the different stages of the vaccination campaign. The lack of information about vaccines and personal belief on vaccine safety and adverse effects were the main reasons for vaccine refusal [16]. Since the vaccination campaign progress and regulatory measures could have changed the HCWs’ inclination to accept or refuse the vaccine, a survey carried out after the immunization program could help to better explain the COVID-19 vaccine acceptance predictors.

Thus, to analyze the reasons that may have led to individual refusal or postponement of vaccination among Italian HCWs, we carried out an online survey to investigate the prevalence rate and the main predictors of vaccine uptake among vaccinated operators in relation to different levels of hesitancy, according to the SAGE group theorical model. Understanding the determinants of accepting or refusing the COVID-19 vaccination is crucial for planning specific interventions to increase the rate of vaccine coverage among healthcare professionals.

## 2. Materials and Methods

### 2.1. Study Design

We conducted a cross-sectional survey, which was administered online to easily gather information on potential factors related to vaccine acceptance or hesitancy. The questionnaire was administered on HCWs in eighteen Italian regions (Abruzzo, Basilicata, Calabria, Campania, Emilia Romagna, Friuli Venezia Giulia, Lazio, Liguria, Lombardia, Marche, Molise, Piemonte, Puglia, Sardegna, Sicilia, Toscana, Valle d’Aosta, and Veneto), selected on the basis of proximity by the collaborating authors, between 30 June and 4 July 2021.

### 2.2. Data Collection

The data were collected by seven institutions: the Department of Medicine and Health Sciences of the University of Molise; the Department of Occupational Medicine of the University of Tor Vergata in Rome; CEIS EEHTA, DEF Department of Faculty of Economics of University of Rome Tor Vergata; the Department of Medicine and Science of Ageing; the Department of Neuroscience, University of Padua; the Institute of Public Health, Section of Legal Medicine of the Catholic University of Sacred Heart of Rome; and the Office of Medical Forensic Coordination, Italian National Social Security Institute (INPS).

Informed consent was obtained at the beginning of the questionnaire, before answering the questions. For the data collection, we used a validated questionnaire that collected data on factors related to the acceptance of the vaccine according to the SAGE group theorical model on vaccine hesitancy [6]. The questionnaire was previously validated in published studies on the same populations, and was modeled on a validated Italian Institute of Health’s questionnaire regarding the psychological impact of COVID-19 infection on the Italian population [17].

The questionnaire was administered via LimeSurvey^©^, a platform which facilitates online surveys and questionnaires. The platform enables the use of different types of questions, with the option to insert conditions and hierarchical dependencies between the questions themselves. Once created, the surveys were activated and distributed through the unique link of the questionnaire, which can be public or with restricted access.

In this specific case, the questionnaire was public and distributed using the mailing lists of the health personnel, starting from the HCWs of the structures that carried out the study. All of the HCWs who responded to the questionnaire were included in the study, and no exclusion criteria were established. We used the Raosoft software (Raosoft, Inc. 6645 NE Windermere Road, Seattle, WA, USA) to estimate the sample size. Keeping the margin of error at 5%, the confidence interval at 95% and the population size at 5000, the sample size was calculated as larger than 350. According to data from the literature that estimated that the average of respondents should be about 5–6%, we can suppose that a large percentage of subjects who opened the link of the survey fulfilled the questionnaire [18]. This distribution method allowed the rapid collection of a total of 757 questionnaires, which were subsequently subjected to statistical and descriptive analyses.

We also asked:-if they would accept the vaccination again;-if they have had any side effects after vaccination, and if these symptoms have led to the loss of working days.

We also asked:-if they believe vaccination is the best way to achieve an immune response;-if the vaccines currently available are safe and effective;-if the vaccination policies (currently decided by the government) are in their best interest;-if vaccines could have adverse effects on the immune system;-if the information received on vaccines and their safety was complete and reassuring;-if the authorities maintained a transparent attitude regarding the possible side effects of vaccination;-if it is important to be vaccinated to protect people who cannot be;-if information heard/read in the media or on social networks influenced their choice;-if they have been motivated to get vaccinated by previous vaccination experience;-whether they trust in the information received regarding the vaccine;-if they are aware that many of the vaccine-preventable diseases are serious;-whether the possibility of vaccination side effects made them question their decision;-whether they think the new vaccines have been tested to the same rigorous standard that normally applies to various drugs;-if they have been vaccinated by law;-if they are afraid of losing their job or of being suspended without pay in case of refusal.

Notably, subjects who refused vaccination were excluded from further analysis, as the study focused on vaccinated HCWs. According to the SAGE model, the survey investigated factors related to vaccine hesitancy: contextual (demographic, socioeconomic), group factors, and factors related to specific COVID-19 hesitancy, concern about infection, and vaccine-related behaviors and intentions [6].

### 2.3. Statistical Analysis

All data were processed using Stata 21.0 software.

Demographics (age group, gender, and region) and work-related factors (job task, high risk setting, and work area) were reported as descriptive statistics. Quantitative data are reported as mean ± SD (standard deviation). Categorical variables are indicated as number (percentage) of participants.

We analyzed the determinants of COVID-19 vaccine acceptance through a two-way cross-table (Fisher’s Exact Test) between all possible hesitation factors and intention/acceptance of the COVID-19 vaccine.

We performed a logistic regression analysis to explore the association between hesitation for COVID-19 vaccination and all factors collected after age, gender, and job task adjustment. Variables were selected only if related to the major outcome in univariate analysis, and were considered in the final model. All *p*-values were two-tailed, and we set the significance level at 5%. To evaluate the degree of hesitancy, we asked the interviewed operators how many days it took them to decide whether to be vaccinated or not.

To evaluate the main determinants of vaccine hesitancy among the study population, we constructed a two-way table evaluating the association of all study variables with vaccine acceptance behaviors (vaccine accepted as soon as it was offered, it took less than a week to decide, it took more than a week to decide, or vaccine rejected). We assumed that the longer it took to decide, the greater the level of hesitancy. We established three levels of hesitancy: non-hesitancy (subjects who agreed to be vaccinated “as soon as they were offered the vaccine”), low-grade hesitancy (operators who declared “to have waited less than one week”) and high-grade hesitancy (those who took “more than one week” before getting vaccinated).

## 3. Results

### 3.1. Demographic and Job Characteristics

Table 1 and Table 2 show the main results of the questionnaire regarding hesitancy among our study population. A total of 757 HCWs completed the online questionnaire. According to the demographic composition of the HCWs population [19], most of the responders were female (511; 67.5%), while the male gender was less represented (246; 32.5%). The median age was 40.7 years ± 13.9. Most of the study population was between 26–45 years of age (359, 47.4%) and 46–67 (265, 35.1%), whereas the 18–25 and >67 age groups were less represented (116, 15.3% and 17, 2.2%, respectively), according to the healthcare sector’s demographic composition.

Regarding job tasks, 320 were medical doctors (42.3%), 146 were nurses (19.3%), 91 were students (12.0%), 147 were psychologists (19.4%), and 53 were other HCWs (7.0%). We found that 38.3% of operators who filled out the questionnaire were employed in a hospital setting, while 12.5% worked in a private clinic, 5.5% in nursing homes or home care, and 1.8% in COVID-19 departments.

### 3.2. Results of the Questionnaire

Most of the study population reported (Figure 1) being vaccinated (730/757; 96.4%), while 27 out of 757 (3.6%) were not; those unvaccinated subjects were excluded from further statistical analysis. To assess the hesitation among the vaccinated, we asked the vaccinated if they had delayed their vaccination or not. Most of the interviewed (88.2%) replied that they were vaccinated “as soon as they were offered the vaccine”, while 1.6% said that they waited less than one week, and 6.5% said that it took more than a week before agreeing to be vaccinated.

Most of our sample (84.9%) stated that they did not regret being vaccinated. To the question of whether they would accept vaccination again, only 4.1% of the study population answered that they would no longer accept it.

Side effects were generally reported (283 out of 757, 37.4% of participants) after vaccination. Among the operators who experienced symptoms related to vaccination, 93 out of 283 subjects reported effects that did not lead to the loss of working days, 137 out of 283 remained at home for 1–2 days, while 45 out of 273 needed to stay at home for 3 to 5 days due to vaccine side effects.

Regarding vaccine beliefs, 97.4% of interviewed HCWs believed that vaccination is the best way to achieve an immune response, and 74.9% considered currently available COVID-19 vaccines to be sufficiently safe and effective. Only 9.8% feared the vaccine’s adverse effects on the immune system. Regarding political beliefs, 82.0% of respondents felt that current vaccination policies have been decided by health authorities in their best interest. With regard to institutional communications, 52.6% of the interviewees considered the information received on the safety of vaccines to be complete and reassuring, 77.1% trusted the information received regarding the vaccine, and 55.9% of respondents believed that the authorities adopted a transparent attitude regarding all the possible side effects of vaccination.

Regarding vaccine habits, 87.5% of study subjects were aware that many of the vaccine-preventable diseases are serious, and 80.1% of them reported that previous vaccination experiences positively influenced their willingness to accept the vaccine. Regarding the COVID-19 vaccine approval course, 73.0% of the population surveyed believed it had been tested to the same rigorous standard that normally applies to various drugs and other vaccines.

Only 13.6% of responding HCWs admitted that the information they heard or read on the media or on social networks influenced their decision to be vaccinated, while 15.9% of them replied that the possibility of developing side effects from vaccination influenced their decision to get vaccinated.

Almost all subjects (96.7%) believed that being vaccinated is an important requirement in order to protect people who cannot undergo vaccination for health reasons, or who cannot obtain vaccine immunity due to immunological conditions.

We then evaluated the impact of mandatory vaccination in influencing the vaccination attitude of the HCWs interviewed; 12.1% of the operators who answered the questionnaire admitted to having been vaccinated by law, but this percentage was higher in subjects classified as “hesitant” (25.0% and 44.2% in HCWs who waited less or more than one week, respectively, *p* < 0.05). Moreover, 7.2% of individuals said they had undergone vaccination for fear of losing their job or being suspended by work without pay in case of refusal.

### 3.3. Factors Associated with Vaccine Hesitancy

According to Fisher’s test, hesitancy was statistically related to work as a nurse, work in high-risk settings (COVID-19 areas and hospital), female gender, distrust in safety and efficacy of vaccines, influence of media or social networks, doubts in the government and vaccination policies, and previous negative experiences with other vaccines. Additionally, we found an association between reduced vaccine uptake and beliefs about the adverse vaccines effects on the immune system, complaints about incomplete information on vaccines safety, and a non-transparent attitude about their possible side effects. Hesitant HCWs most frequently did not believe that vaccination is an important measure to protect people who cannot be immunized. They also believed that most vaccine-preventable diseases are mild, and that the new COVID-19 vaccines had not been tested with the same standard as other drugs or vaccines. Legal obligation and fear of losing their job or being suspended without pay in case of refusal were strongly associated with the level of vaccine hesitancy in our sample.

We then performed a regression analysis to exclude all possible confounding factors. At univariate analysis, after controlling for age and gender, we categorized vaccine hesitancy as the dependent variable and all factors associated with vaccine acceptance as independent variables. We considered all subjects who received the vaccine as soon as possible to be “not hesitant”, and all the others to be “hesitant”. We found that in our HCWs population, vaccine hesitancy was significantly positively associated with fear of vaccination side effects (*p* < 0.01), and negatively related to confidence in the safety and efficacy of the vaccine (*p* < 0.01). After performing multivariate analysis (Table 3), only the fear of possible vaccination side effects (OR: 4.631, *p* < 0.01) and the confidence in vaccine safety and effectiveness (OR: 0.35, *p* < 0.05) were associated with vaccine hesitancy. Neither mandatory vaccination nor fear of job loss due to vaccine refusal were found to be significantly related to vaccine hesitancy by the multivariate analysis.

## 4. Discussion

Vaccine hesitancy in HCWs is a complex phenomenon. Improving understanding of the major determinants of vaccine uptake among these individuals is crucial. The reasons for vaccine hesitancy among HCWs are complex and varied, as the term “healthcare workers” itself applies to a group of workers with different job tasks and socioeconomic classes, as well as heterogeneous cultural levels. For many months, the only reliable measure to mitigate the impact of COVID-19 pandemic was to reduce the frequency of interpersonal contact and the use of PPE. Social distancing can effectively reduce the spread of SARS-COV2 in the community, but it imposes high social costs due to reduced economic activities, and thus, vaccine development was accelerated around the world. Although vaccination of HCWs was a priority, a suboptimal rate of vaccine acceptance and a high rate of hesitancy were observed among these workers globally. Our study shows a mixed picture of vaccination habits among HCWs, and adds substantial knowledge of how the main study factors influenced hesitant operators. We found an overall vaccine uptake rate of 96.4% in our sample. Acceptance was significantly related to job task, with physicians showing highest rate of uptake compared to other occupations (see Table 4). The overall acceptance rate was higher than estimated in previous studies surveys based on vaccine intention [13,14]. Vaccine intention was higher in male subjects, the elderly, and medical doctors, while female subjects, younger workers, nurses, and auxiliary health personnel showed higher rates of hesitation. In previous studies, the vaccine acceptance rate was related to previous vaccination habits and confidence in vaccine efficacy rather than fear of the negative health consequences of vaccination [20,21,22,23].

In a recent scoping review, intention to be vaccinated was reported to range between 27 and 77% among HCWs. In a study carried out on Chinese nurses, only 40% of operators were prone to accepting the COVID-19 vaccine, while a higher vaccine acceptance rate were reported in surveys from Israel and France (77 and 78%, respectively). In a study conducted in Canada, the vast majority of interviewed healthcare workers reported willingness to be vaccinated (around 80%) [24]. The rates of vaccine acceptance among HCWs in South America were reported to be high (77% in Colombia [25] and 80.7% in Brazil [26]). Data from a South African survey showed a rate of 91% vaccine acceptance among healthcare workers [27], which is quite similar to our findings. Data from Germany reported that 66% of HCWs were still vaccinated in January 2020, and that 22% had planned to receive the vaccine, yielding a low rate of hesitancy among this population [28]. Findings from a large UK HCWs cohort reported 23% of them being vaccine hesitant, with ethnic minority status being significantly associated with hesitancy [14]. In a survey among 1723 Italian HCW, an overall intention of 67% to be vaccinated was found, while 26% were not sure, and 7% said that they would refuse the vaccine [29]. In a previous survey, we found a COVID-19 vaccination acceptance rate of 91.5% among nurses from four Italian regions at the beginning of the vaccination campaign [20]. Geographical and temporal variations in vaccination aptitude for COVID-19 are widely reported in the literature, but comparisons between studies from different countries and time periods are difficult to carry out.

Low risk perception, lack of use of evidence based information sources, and low confidence in scientists and in the effectiveness of government measures (including conspiracy theories on secret agreements between pharmaceutical companies and the Italian government) have been recognized as possibly responsible for low vaccine uptake among HCWs [30,31]. These findings are confirmed in the present study by the significant percentage of operators that showed a lack of trust in government policies (19.4%) and did not believe that vaccine strategies were decided in their best interest by stakeholders (15.2%). This is outlined by the high percentage (37.4%) of respondents who considered the authorities’ attitude regarding the possible side effects of vaccination not to be transparent. Moreover, 11.5% of our population affirmed that their decision on vaccine acceptance was influenced by social media. In a recent Italian study, the frequency of media use was related to the level of conspiracy-mindedness [32]. Studies on the influence of income and education on vaccine intention have shown conflicting results [33,34,35]; our study failed to show any difference in vaccine acceptance among various healthcare job tasks.

Most of the studies mentioned above have focused on vaccine attitude rather than vaccine adoption among healthcare workers, so it is difficult to compare these findings with our study results; moreover, it can be assumed that a variable percentage of hesitant individuals found in those surveys decided to get vaccinated later. Italy was one the earliest and most affected countries at the start of the COVID-19 pandemic, and recorded a high death rate among western European countries. According to the health belief model, we expected and found a high vaccine acceptance rate among Italian operators. The results of our survey clearly show that the SAGE model explaining vaccine hesitancy among general population is also well suited to healthcare professionals. In our survey, upon univariate analysis, nearly all investigated factors were significantly correlated with vaccine hesitancy among operators. Lack of trust in rulers and health systems, belief in non-transparent reporting of adverse vaccine effects by health governance and pharmaceutical companies, and fear of vaccination side effects were the key factors associated with a higher level of hesitancy in our sample, and the efficacy and safety of vaccines were significantly related to hesitancy.

Previous international and Italian experiences were mainly based on influenza and H1N1 vaccination, and the highlights, based on the health belief model (HBM), have shown how the intention to be vaccinated is related to the following factors: the perception that the vaccine is safe and effective in protecting ourselves and our relatives; the disease is perceived as serious and severe; easy access to scientific literature; trust in public communicators; and encouragement from relatives and colleagues [36]. Since, according to the SAGE group’s statement, vaccine attitude is a continuum, ranging from total acceptance to complete refusal, and hesitant operators are a heterogeneous group in the middle of this continuum, we expected mandatory vaccination policies to be more effective on individuals who were less hesitant than others. Thus, upon univariate analysis, the rate of subjects who accepted vaccination “by law” and for fear of losing their own jobs in case of vaccine refusal was found to be significantly related to, and showed a clear positive trend with, growing levels of hesitancy. However, upon logistic regression analysis, the outcome of “hesitation” was not found to be significant after controlling for other relevant variables in the model.

Upon multivariate analysis, as shown in Table 3, we found that the only factors that remained significantly related to vaccine hesitancy were the fear of possible vaccination side effects (OR: 4.631, *p* < 0.01) and the confidence in vaccine safety and effectiveness (OR: 0.35, *p* < 0.05). These findings confirm results from previous studies, in which confidence in vaccine efficacy was shown to be the best predictor of vaccine acceptance rate among nurses (intention to be vaccinated), and that interventions aimed at improving workers’ confidence in vaccine efficacy at the population and individual levels can help to achieve a high rate of vaccine acceptance rate among health operators. According to WHO, vaccine safety concerns are associated with a high degree of vaccine hesitation, but it is important not to equate vaccine hesitancy and vaccine safety, as the latter is only one of the vaccine hesitancy factors [5,6]. Surprisingly, in our sample, while both are associated with the degree of hesitation on univariate analysis, neither “by law” vaccination nor the fear of losing one’s own job due to vaccine refusal was found to be significantly related to vaccine hesitancy after controlling for relevant determinants. This result could simply be attributable to the relatively low number of hesitant subjects in the model, and to the high number of variables considered that failed to reach statistical significance. However, this result is notable, since mandatory vaccination is considered the most effective intervention to induce a high rate of vaccination in the general population [37,38]. We can suppose that anti-vaccine beliefs in healthcare workers are widely based on personality [14,39], and are, thus, deeply rooted and independent of personal and medical knowledge. On this basis, those beliefs are not accessible to common judgement and are poorly influenced by rules of law; past experiences reported that the greatest difficulties encountered in convincing hesitant subjects are represented by their scarce willingness to talk about their own beliefs. Such factors could be particularly relevant in highly skilled practitioners such as HCWs.

One of the limits of the study is that we could not prevent double or multiple entries during the online survey, but given the relatively long time requested to fulfill the questionnaire, this eventuality is not likely. Another possible limit of the study is that the Italian regions sampled were selected based on the location of the centers that collaborated in the study, so we cannot exclude possible bias due to the different levels of hesitancy among the various regions. However, data on vaccine coverage of the study period showed similar rates of vaccine uptake in Italy, so regional differences are unlikely to be present. Indeed, the rate of hesitancy between different regions tested as not significant, as shown in Table 1 and Table 2.

## 5. Conclusions

Our study showed a 96.4% rate of vaccine acceptance among Italian HCWs and a strong association between hesitancy and personal beliefs. A high percentage of the study population showed lack of trust in government policies and high social media influence on vaccine acceptance. The main predictors of COVID-19 vaccination uptake among Italian HCWs were found to be the fear of possible vaccination side effects and the level of confidence in vaccine safety and effectiveness. Action to improve operators’ trust in institutions and vaccine safety could result in a wider acceptance rate among those subjects.

## Figures and Tables

**Figure 1 tropicalmed-07-00419-f001:**
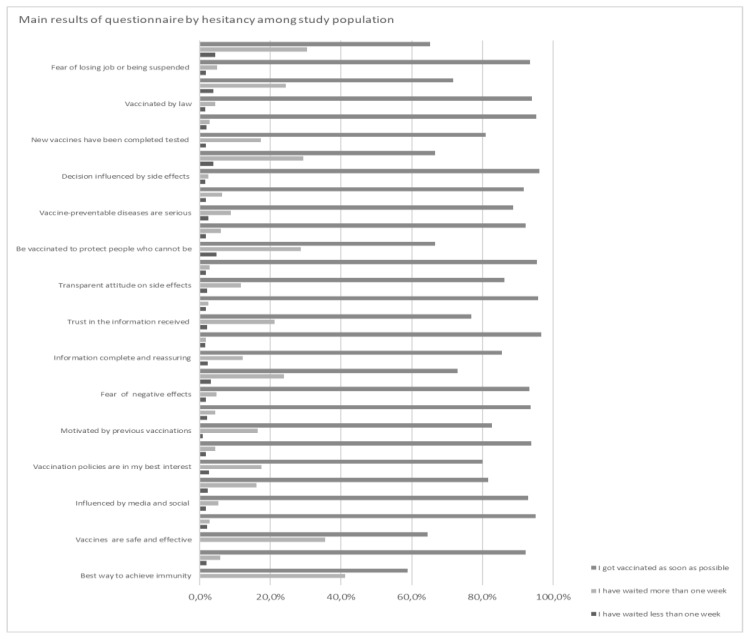
Questionnaire answers by hesitancy levels.

**Table 1 tropicalmed-07-00419-t001:** Main results of the questionnaire on factors related to vaccine acceptance by different levels of hesitancy among the study population.

		Did You Get Vaccinated as Soon as You Were Offered the Vaccine or did You Take Some Time to Think about it?						
		I Have Waited Less Than One Week		I Have Waited More Than One Week		I Got Vaccinated as Soon as Possible		
		Count	Row Valid N %	Count	Row Valid N %	Count	Row Valid N %	*p* Value
Vaccination is the best way to achieve an immune response	No	0	0.00%	7	41.20%	10	58.80%	*p* < 0.01
	Yes	12	1.90%	36	5.80%	578	92.30%	
Vaccines currently available are safe and effective	No	0	0.00%	27	35.50%	49	64.50%	*p* < 0.01
	Yes	12	2.10%	16	2.80%	539	95.10%	
The information heard/read on the media or on social networks influenced the choice	No	10	1.80%	29	5.20%	516	93.00%	*p* < 0.01
	Yes	2	2.30%	14	16.10%	71	81.60%	
The vaccination policies currently decided by the government are in my best interest	No	3	2.60%	20	17.40%	92	80.00%	*p* < 0.01
	Yes	9	1.70%	23	4.40%	496	93.90%	
Motivated to get vaccinated by previous experiences with vaccinations	No	1	0.80%	21	16.40%	106	82.80%	*p* < 0.01
	Yes	11	2.10%	22	4.30%	482	93.60%	
Vaccines can have negative effects on the immune system	No	10	1.70%	28	4.80%	540	93.40%	*p* < 0.01
	Yes	2	3.20%	15	23.80%	46	73.00%	
The information received on vaccines and their safety was complete and reassuring	No	7	2.30%	37	12.20%	260	85.50%	*p* < 0.01
	Yes	5	1.50%	6	1.80%	327	96.70%	
Trust in the information received regarding the vaccine	No	3	2.00%	31	21.10%	113	76.90%	*p* < 0.01
	Yes	9	1.80%	12	2.40%	474	95.80%	
The authorities have maintained a transparent attitude regarding the possible side effects of vaccination	No	6	2.10%	33	11.70%	244	86.20%	*p* < 0.01
	Yes	6	1.70%	10	2.80%	342	95.50%	
It is important to be vaccinated to protect people who cannot be	No	1	4.80%	6	28.60%	14	66.70%	*p* < 0.01
	Yes	11	1.80%	37	6.00%	573	92.30%	
Aware that many of the vaccine-preventable diseases are serious	No	2	2.50%	7	8.80%	71	88.80%	*p* < 0.05
	Yes	10	1.80%	36	6.40%	515	91.80%	
The possibility of side effects from vaccination made you question your decision	No	8	1.50%	13	2.40%	518	96.10%	*p* < 0.01
	Yes	4	3.90%	30	29.40%	68	66.70%	
Think that the new vaccines have been tested with the same rigorous standard that normally applies to various drugs	No	3	1.70%	30	17.30%	140	80.90%	*p* < 0.01
	Yes	9	1.90%	13	2.80%	446	95.30%	
Vaccinated by law	No	9	1.60%	24	4.30%	531	94.10%	*p* < 0.01
	Yes	3	3.80%	19	24.40%	56	71.80%	
Fear of losing job or being suspended without salary in case of refusal	No	10	1.70%	29	4.90%	557	93.50%	*p* < 0.01
	Yes	2	4.30%	14	30.40%	30	65.20%	

**Table 2 tropicalmed-07-00419-t002:** Association between different level of vaccine acceptance and demographic/occupational characteristics of the study population.

		Did You Get Vaccinated as Soon as you Were Offered the Vaccine or did You Take Some Time to Think about it?						
		I Have Waited Less Than One Week		I Have Waited More Than One Week		I Got Vaccinated as Soon as Possible		
		Count	Row Valid N %	Count	Row Valid N %	Count	Row Valid N %	*p* Value
Gender	Female	9	1.80%	35	6.80%	443	86.70%	*p* < 0.05
	Male	3	1.20%	14	5.70%	225	91.50%	
Age	>45	5	1.80%	13	4.70%	249	89.20%	*p* < 0.05
	<45	7	1.50%	36	7.60%	416	87.60%	
Nurse	No	8	1.30%	41	6.70%	539	88.20%	*p* < 0.05
	Yes	4	2.70%	8	5.50%	129	88.40%	
High risk setting	No	0	0.00%	0	0.00%	8	1.80%	*p* < 0.05
	Yes	0	0.00%	0	0.00%	11	3.60%	
Working area	Clinic	1	1.10%	6	6.30%	85	89.50%	*p* = 0.29
	Home Assistance	1	4.50%	3	13.60%	17	77.30%	
	Nursing Homes	0	0.00%	3	15.00%	17	85.00%	
	Hospital	4	1.40%	13	4.50%	267	92.10%	
	COVID-19 areas	0	0.00%	0	0.00%	14	100.00%	
	Other	6	1.90%	24	7.60%	268	84.80%	
Regions	Abruzzo	0	0.00%	0	0.00%	5	100.00%	*p* = 0.73
	Sicilia	0	0.00%	0	0.00%	4	100.00%	
	Basilicata	0	0.00%	0	0.00%	15	100.00%	
	Calabria	0	0.00%	1	16.70%	5	83.30%	
	Campania	2	1.00%	18	8.90%	172	85.10%	
	Emilia Romagna	0	0.00%	2	14.30%	11	78.60%	
	Friuli Venezia Giulia	0	0.00%	2	13.30%	13	86.70%	
	Lazio	5	2.50%	12	5.90%	185	90.70%	
	Liguria	0	0.00%	0	0.00%	6	100.00%	
	Lombardia	0	0.00%	3	10.00%	24	80.00%	
	Marche	0	0.00%	0	0.00%	2	100.00%	
	Molise	2	1.20%	5	3.00%	150	88.80%	
	Piemonte	1	11.10%	0	0.00%	8	88.90%	
	Puglia	0	0.00%	0	0.00%	7	100.00%	
	Sardegna	1	6.30%	2	12.50%	13	81.30%	
	Toscana	1	2.70%	2	5.40%	34	91.90%	
	Valle d’Aosta	0	0.00%	0	0.00%	1	100.00%	
	Veneto	0	0.00%	2	13.30%	13	86.70%	
Profession	Nurse	4	2.70%	8	5.50%	129	88.40%	*p* < 0.05
	Medical doctor	6	1.90%	17	5.30%	290	90.60%	
	Psychologist	0	0.00%	17	11.60%	118	80.30%	
	Student	0	0.00%	4	4.40%	86	94.50%	
	Other HCWs	2	3.80%	3	5.70%	45	84.90%	

**Table 3 tropicalmed-07-00419-t003:** Predictors influencing participants’ vaccine intention (multivariate analysis).

	B	S.E.	Wald	df	Sig.	Exp(B)	95% C.I. for EXP(B)	
							Lower	Upper
Age < 45 years	0.244	0.384	0.403	1	0.526	10.276	0.244	0.384
Male gender	−0.028	0.396	0.005	1	0.944	0.973	−0.028	0.396
Vaccination is the best way to achieve an immune response	0.268	0.680	0.156	1	0.693	10.308	0.268	0.680
Vaccines currently available are safe and effective	−10.075	0.476	50.095	1	0.024	0.341	−10.075	0.476
The information heard/read on the media or on social networks influenced the choice	0.448	0.394	10.288	1	0.256	10.565	0.448	0.394
The vaccination policies currently decided by the government are in my best interest	0.004	0.426	0.000	1	0.993	10.004	0.004	0.426
Motivated to get vaccinated by previous experiences with vaccinations	−0.015	0.395	0.001	1	0.969	0.985	−0.015	0.395
Vaccines can have negative effects on the immune system	0.399	0.440	0.823	1	0.364	10.490	0.399	0.440
The information received on vaccines and their safety was complete and reassuring	−0.253	0.489	0.267	1	0.605	0.777	−0.253	0.489
Trust in the information received regarding the vaccine	−0.295	0.492	0.359	1	0.549	0.745	−0.295	0.492
The authorities have maintained a transparent attitude regarding the possible side effects of vaccination	−0.222	0.404	0.301	1	0.583	0.801	−0.222	0.404
It is important to be vaccinated to protect people who cannot be	−0.848	0.672	10.589	1	0.207	0.428	−0.848	0.672
Aware that many of the vaccine-preventable diseases are serious	0.348	0.481	0.522	1	0.470	10.416	0.348	0.481
The possibility of side effects from vaccination made you question your decision	10.522	0.405	140.134	1	0.000	40.581	10.522	0.405
Think that the new vaccines have been tested with the same rigorous standard that normally applies to various drugs	0.040	0.435	0.008	1	0.927	10.041	0.040	0.435
Vaccinated by law	0.444	0.448	0.982	1	0.322	10.559	0.444	0.448
Fear of losing job or being suspended without salary in case of refusal	0.236	0.510	0.214	1	0.644	10.266	0.236	0.510
Nurse job	0.344	0.436	0.623	1	0.430	10.411	0.344	0.436

**Table 4 tropicalmed-07-00419-t004:** Rate of vaccine uptake by job tasks.

			Profession			Total
			Medical Doctor	Nurse	Other HCWs	
Vaccinated	No	Count	7	5	15	27
		% within profession	2.2%	3.4%	5.2%	3.6%
	Yes	Count	313	141	276	730
		% within profession	97.8%	96.6%	94.8%	96.4%
Total		Count	320	146	291	757
		% within profession	100.0%	100.0%	100.0%	100.0%
**Chi-Square Tests**						
	Value	df	Asymptotic Significance (2-sided)	Exact Sig. (2-sided)	Exact Sig. (1-sided)	Point Probability
Pearson Chi-Square	3.912	2	0.141	0.138		
Likelihood Ratio	3.936	2	0.140	0.133		
Fisher’s Exact Test	3.836			0.132		
Linear-by-Linear Association	3.886	1	0.049	0.050	0.031	0.013
N of Valid Cases	757					

## Data Availability

The data presented in this study are available upon request from the corresponding author. The data are not publicly available for ethical reasons.

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
