# Peer review of "Main Predictors of COVID-19 Vaccination Uptake among Italian Healthcare Workers in Relation to Variable Degrees of Hesitancy: Result from a Cross-Sectional Online Survey"

_tropicalmed, 2022, doi:10.3390/tropicalmed7120419_

Round 1
Reviewer 1 Report
Title:
1. Include the type of study design in the main title.
2. Replace “different levels” by using a synonym.
3. Is it “main determinants” or “predictors”?
Abstract:
1. Use colon after Background, Methods, Results and Conclusion.
2. line 1: replace “biggest” by “major”.
3. Line 1 is unclear. “despite vaccines”? the sentence is incomplete.
4. line 3: “o” to be replaced by “or”
5. line 6: it should be vaccine acceptance “and” not “or” hesitancy
Keywords: change the keywords as they should not be identical with the title words
Introduction and discussion: Needs to be strengthened by including the following points:
1. Include information about COVID-19 origin and transmission.
2. Define vaccine hesitancy and resistance.
3. Mention the main predictors of vaccine hesitancy on a global scale (refer and cite: doi: 10.1136/postgradmedj-2021-141365).
4. Include common treatment options for COVID-19 apart from preventive vaccination measures.
5. Compare the role of health care officials and government organizations in Italy in combatting COVID-19 from other regions (refer and cite: doi: 10.3389/fpubh.2022.844333).
Materials and methods:
1. Indicate the reasons why cross sectional survey was a preferred mode.
2. Specify the 17 regions where the survey was conducted.
3. Indicate the measures taken by the research group to validate the survey questionnaire. Where there any particular variables that were selected during this validation.
4. During statistical evaluation, indicate if any bias was generated. If YES indicate the confounding factors and the measures to minimize the bias.
5. Provide “inclusion and exclusion” criteria during selecting the survey population.
6. The authors fail to address if there were any vaccine resistant cases during their survey. Including vaccine resistant” participants as high grade vaccine hesitancy” would cause a serious bias in the survey analysis. This needs to be clarified in detail.
7. Elaborate the “descriptive analysis” undertaken by the authors.
Results:
1. Why was a gender bias generated in the study? Was the survey population streamlined towards women professions? This needs explanation.
2. if the age group 18-25 and >67years were under-represented, why was this group excluded from the study?
3. The 1.8% of the survey population were in COVID-19 departments, does this include the hospital staff or were there just non-medical staff members? This would create a bias in KAP mode (Knowledge, attitude and practice).
Figures and Table:
1. Include few graphical representations of the tables.
Author Response
We thanks the reviewer for his valuable suggestions that greatly implemented the significance and readeablity of the study. We have made all the suggested changes.
Title:
- Include the type of study design in the main title.
- Replace “different levels” by using a synonym.
- Is it “main determinants” or “predictors”?
1.2.3. We changed the title as suggested.
Abstract:
- Use colon after Background, Methods, Results and Conclusion.
- line 1: replace “biggest” by “major”.
- Line 1 is unclear. “despite vaccines”? the sentence is incomplete.
- line 3: “o” to be replaced by “or”
- line 6: it should be vaccine acceptance “and” not “or” hesitancy
1.2.3.4.5. We changed the abstract as requested.
Keywords: change the keywords as they should not be identical with the title words
We changed the Keyword has suggested.
Introduction and discussion: Needs to be strengthened by including the following points:
- Include information about COVID-19 origin and transmission.
- Define vaccine hesitancy and resistance.
- Mention the main predictors of vaccine hesitancy on a global scale (refer and cite: doi: 10.1136/postgradmedj-2021-141365).
- Include common treatment options for COVID-19 apart from preventive vaccination measures.
- Compare the role of health care officials and government organizations in Italy in combatting COVID-19 from other regions (refer and cite: doi: 10.3389/fpubh.2022.844333).
1.2.3.4.5. We changed the Introduction and discussion including the suggested definitions and citations.
Materials and methods:
- Indicate the reasons why cross sectional survey was a preferred mode.
- Specify the 17 regions where the survey was conducted.
- Indicate the measures taken by the research group to validate the survey questionnaire. Where there any particular variables that were selected during this validation.
1.2.3. We made the opportune change in the text.
- During statistical evaluation, indicate if any bias was generated. If YES indicate the confounding factors and the measures to minimize the bias.
4.There were no bias during the statistical evaluation.
- Provide “inclusion and exclusion” criteria during selecting the survey population.
5.There were no inclusion and exclusion criteria.
- The authors fail to address if there were any vaccine resistant cases during their survey. Including vaccine resistant” participants as high grade vaccine hesitancy” would cause a serious bias in the survey analysis. This needs to be clarified in detail.
- Since the questionnaire was intended only to vaccinated subjects, no resistant subject was included in the “high grade vaccine hesitancy” group.
- Elaborate the “descriptive analysis” undertaken by the authors.
- We elaborated the descriptive analysis as requested.
Results:
- Why was a gender bias generated in the study? Was the survey population streamlined towards women professions? This needs explanation.
- We made the opportune explanation in the text.
- if the age group 18-25 and >67years were under-represented, why was this group excluded from the study?
- The age group 18-25 and >67years were under-represented, according to the healthcare sectors demographic composition. However, in order to control possible age bias, age was included in the logistic regression model.
- The 1.8% of the survey population were in COVID-19 departments, does this include the hospital staff or were there just non-medical staff members? This would create a bias in KAP mode (Knowledge, attitude and practice).
- Only HCWs were included in the study.
Figures and Table:
- Include few graphical representations of the tables.
- A graphic on questionnaire answers by hesitancy levels was added.
Reviewer 2 Report
Explain the results of mulivariate analysis and refer to which it table ?
Table 3, please interprete the results
Do not copy analysis results , create on your own style.
Table 2-4 , Correct typo errors comma" , " to dot ". "
Author Response
We are very grateful to the reviewer for his suggestions that deeply implemented the significance and understanding of the study. We made all the requested changes.
Explain the results of mulivariate analysis and refer to which it table ?
We add the explanation in the results section.
Table 3, please interprete the results
Done.
Do not copy analysis results , create on your own style.
Graphical explanation was added to the results section. (Graphic 1)
Table 2-4 , Correct typo errors comma" , " to dot ". "
Done.
Reviewer 3 Report
please see the attached file.

Author Response
We thank the reviewer for his invaluable suggestion and the criticisms that have deeply implemented the meaning, relevance and readability of the study. We have made all the suggested changes and improved the various sections with the required insights and clarifications.
Comments to authors
The manuscript may be of some value particularly to the Italian population and healthcare system. However, there are serious methodological flaws, of which some may be irredeemable. Additionally, the authors did not extensively discuss possible reasons, implications and ways-outs for the suboptimal rate of COVID-19 vaccine acceptance among Italian HCWs. In my opinion, these should be the crux of this manuscript to highlight the public health significance and help the paper to attain a publishable value. English language inaccuracies were rampant and a moderate English language editing may be required.
Title: Consider this: “Main determinants of COVID-19 vaccination uptake and levels of the vaccine hesitancy among Italian healthcare workers”
Abstract:
- The abstract is too scanty. Main findings, e.g. determinants of COVID-19 vaccine update and the levels of the vaccine acceptance according to various variables were not presented. Possible reasons, implications and ways-outs for the suboptimal rate of COVID-19 vaccine acceptance among Italian HCWs found in the study is not highlighted. The abstract shouldn’t necessarily be structured but need to be populated with facts and figures (values or proportions) found. The concise implication of the findings of the study (major take homes) to the Italian population and healthcare system needs to be stated
Abstract has been implemented with most relevant literature findings regarding determinants of vaccine hesitancy among HCWs. We added also the main findings of the study.
Introduction
- The introduction needs to provide a sufficient background to the title and clearly state the study objective as well as the knowledge gaps.
The introduction section has been implemented as requested.
- From the last paragraph of the introduction, the authors “carried on an online survey to investigate the rate and the main determinants of vaccine uptake among vaccinated Italian HCWs in relation to different levels of hesitancy” How can the authors determine the rate of vaccination and or level of hesitance among already vaccinated HCWs? Again, the “level of hesitance” the authors want to or determined, is it according to their own definition or is there a standard level of hesitance that guided the authors?. If the latter is the case, then there is need for a citation.
Explanation of the hesitancy definition used for the study has been added both in the introduction and in the method sections.
Materials and method
- The authors stated that the questionnaire used was validated. Which method of validation was used and how was it performed? Was the questionnaire pilot-tested?
Details about validation of the questionnaire has been added in the text.
- How did the authors arrive at the sample size of 757 respondents surveyed? Sample sizes are calculated based on known prevalence or you assume 50% prevalence when the prevalence is not known. Was this basic epidemiological procedure followed? Alternatively, Sample size can be determined based on a representative proportion or certain percentage (quota) of the number of the target population (in this case the number of HCWs in the selected Italian regions). How did the authors determine their sample size?
Explanation on the sample sizing has been added in the opportune section of the paper.
- How did the author select the Italian regions surveyed to convince the readership that there was no bias? Was it purposive selection? If, then give reason for the adoption of that method.
We specified how we selected the Italian regions surveyed in the discussion.
- During the online survey, how did the authors prevent double or multiple entries?
We could not prevent it, so we specified it in the discussion, as a limit of the study.
- What were the inclusion criteria for selection of respondents surveyed?
Inclusion criteria were added in the text.
Results
- Parts of the results read as if it is materials and methods. Examples Lines 178-179, 185-188, 201, et c. The authors can explain these in the M&M and then just present their results without referring to how and why it was done.
Requested changes have been done.
- Table 1. What does “n.s” stand for under the column p-value? Not significant? I think the authors should just present the exact p-values computed for each of the variable/factors and allow the readership to determine if the p-value is significant or not; having defined or set the statistical significant p-value in the M&M.
We added p-values in the table.
- Table 1 is too long in the view of this reviewer. Ù
For a better comprehension we split the table 1 in two different tables and then we added graphic 1.
- The table titled are two scanty and unable to stand alone. The readership should be able to read the table title and make sense out of it without refereeing to the whole paper.
We modified the table titles as requested.
- Table 3 should be mentioned and presented in the result section and not in the discussion.
Done.
Discussion
- The possible reasons, implications and ways-outs for the suboptimal rate of COVID-19 vaccine acceptance among Italian HCWs should be the crux of the discussion and conclusion. Without this, the manuscript is not worth publication in Tropical Medicine and Infectious Disease, in my view.
The discussion and conclusion sections have been implemented as suggested.
Reviewer 4 Report
Comments to authors
The manuscript may be of some value particularly to the Italian population and healthcare system. However, there are serious methodological flaws, of which some may be irredeemable. Additionally, the authors did not extensively discuss possible reasons, implications and ways-outs for the suboptimal rate of COVID-19 vaccine acceptance among Italian HCWs. In my opinion, these should be the crux of this manuscript to highlight the public health significance and help the paper to attain a publishable value. English language inaccuracies were rampant and a moderate English language editing may be required.
Title: Consider this: “Main determinants of COVID-19 vaccination uptake and levels of the vaccine hesitancy among Italian healthcare workers”
Abstract:
- The abstract is too scanty. Main findings, e.g. determinants of COVID-19 vaccine update and the levels of the vaccine acceptance according to various variables were not presented. Possible reasons, implications and ways-outs for the suboptimal rate of COVID-19 vaccine acceptance among Italian HCWs found in the study is not highlighted. The abstract shouldn’t necessarily be structured but need to be populated with facts and figures (values or proportions) found. The concise implication of the findings of the study (major take homes) to the Italian population and healthcare system needs to be stated
Introduction
- The introduction needs to provide a sufficient background to the title and clearly state the study objective as well as the knowledge gaps.
- From the last paragraph of the introduction, the authors “carried on an online survey to investigate the rate and the main determinants of vaccine uptake among vaccinated Italian HCWs in relation to different levels of hesitancy” How can the authors determine the rate of vaccination and or level of hesitance among already vaccinated HCWs? Again, the “level of hesitance” the authors want to or determined, is it according to their own definition or is there a standard level of hesitance that guided the authors?. If the latter is the case, then there is need for a citation.
Materials and method
- The authors stated that the questionnaire used was validated. Which method of validation was used and how was it performed? Was the questionnaire pilot-tested?
- How did the authors arrive at the sample size of 757 respondents surveyed? Sample sizes are calculated based on known prevalence or you assume 50% prevalence when the prevalence is not known. Was this basic epidemiological procedure followed? Alternatively, Sample size can be determined based on a representative proportion or certain percentage (quota) of the number of the target population (in this case the number of HCWs in the selected Italian regions). How did the authors determine their sample size?
- How did the author select the Italian regions surveyed to convince the readership that there was no bias? Was it purposive selection? If, then give reason for the adoption of that method.
- During the online survey, how did the authors prevent double or multiple entries?
- What were the inclusion criteria for selection of respondents surveyed?
Results
- Parts of the results read as if it is materials and methods. Examples Lines 178-179, 185-188, 201, et c. The authors can explain these in the M&M and then just present their results without referring to how and why it was done.
- Table 1. What does “n.s” stand for under the column p-value? Not significant? I think the authors should just present the exact p-values computed for each of the variable/factors and allow the readership to determine if the p-value is significant or not; having defined or set the statistical significant p-value in the M&M.
- Table 1 is too long in the view of this reviwer.
- The table titled are two scanty and unable to stand alone. The readership should be able to read the table title and make sense out of it without refereeing to the whole paper
- Table 3 should be mentioned and presented in the result section and not in the discussion.
Discussion
- The possible reasons, implications and ways-outs for the suboptimal rate of COVID-19 vaccine acceptance among Italian HCWs should be the crux of the discussion and conclusion. Without this, the manuscript is not worth publication in Tropical Medicine and Infectious Disease, in my view.
Author Response

(The authors gave the same response as above.)

Round 2
Reviewer 3 Report
I did not see the author's response to my original comments. I attached my comments here again. If the author has addressed the comments, please send it to me again. Thanks!

Author Response
Dear editor,
Thank you for providing me the opportunity to review the article Main determinants of COVID-19 vaccination uptake among Italian healthcare workers (HCWs) in relation to different levels of hesitancy. The paper addressed an interesting topic, however, it has several major problems in writing, statistical analysis, and interpretation. Please see the comments below:
We thank the reviewer for his invaluable suggestion and the criticisms that have deeply implemented the meaning and relevance of the study. We have made all the suggested changes and improved the various sections with the required insights and clarifications.
Comments
- Title: suggest to use “main associated factors” instead of “main determinants”, as the study used a cross-sectional design which has very limited ability of causal inference. We changed the title and the results section according to your suggestions and to a previous reviewer’s request.
- The introduction is not well organized enough. There are too many paragraphs now. Suggest to reframe it, such as the 1st paragraph demonstrated vaccine hesitancy in general population and HCW and its negative impact; the 2nd paragraph goes to the factors associated with vaccine hesitancy among general population and HCW; the 3rd paragraph introduces the research gap; and the 4th paragraph shows the study objectives. We restructured the introduction section according to your suggestion.
- Line 69-70: the authors indicated that “investigate the rate and the main determinants of vaccine uptake among vaccinated Italian HCWs in relation to different levels of hesitancy”. It is confusing because if the target participants are vaccinated HCW, the prevalence of vaccine uptake should be known as 100%. Also, the outcome should be prevalence but not rate which is an outcome measure considering time and should not be used in cross-sectional study. We thank you for the opportune annotation and we changed the text in the introduction section.
- I suggest to add subheadings in Methods section and Results section for better reading.
- Methods: Please reorganize the order of Methods section. For example, the description of variables’ measures usually precedes the description of statistical analysis. We reorganized the method section as suggested.
- Methods: Please define the inclusion criteria and exclusion criteria of study participants Criteria were added in the text.
- Methods: Please describe how many institutions were included in this study
- Methods: Please describe how to select the study institutions and recruit the participants (Define the sampling method) Institution included in the study were specified.
- Methods: Please describe the procedure of informed consent. We described the informed consent procedure in the methods section.
- Line 100: which specific method was used in the two-way cross-tables analysis? Chi-square test? Fisher’s exact test? We added in the method section the specific method used.
- Line 102-104: Please explain the reasons to adjust for age and gender instead of other socio-demographic factors in logistic regression. According to your request, we have added job task in the multivariate analysis. We didn’t stratified for geographical regions, since there were no literature evidence of different acceptance among different Italian regions, moreover the number of respondents in some region was significantly low compared to others.
- Line 105-123: Please reorganize the paragraph for better reading. I suggest adding subheadings for different variables
- In line 102-104, the authors indicated that “We performed a multinomial logistic regression analysis to explore the association between hesitation for COVID-19 vaccination and all factors collected after age and gender adjustment.” But in line 204-205, the authors dichotomized the dependent variable. It seems contradictory. Thank you for your suggestion, we made the opportune correction in the text.
- In line 129-130, the authors indicated that “Of note subjects who refused vaccination were excluded from the analysis”. But according to Results section, the authors included those who were not vaccinated into analysis and analyzed the determinants of COVID-19 vaccine acceptance. We specified in the text that those subjects were excluded in the analysis.
- Line 148-149: I do not understand the statement that “Most of our sample (643/757: 84.9%) stated that they did not regret being vaccinated and no longer accept vaccination (4.1%).” The statement was rephrased.
- Line 185-200: Did these findings display in Tables or Figures? Yes, we displayed it both in Table and in Graphic.
- Did the authors take the collinearity between independent variables into consideration when performing multivariable regressions? Collinearity was tested in the model.
- I suggest reorganizing the Discussion section to highlight key findings. There are too many paragraphs now, some of which can be merged and simplified.
- Discussion: References are missing in the last sentence of paragraph 4. References were added in the text.
- Discussion: Please describe the limitations of this study
- Conclusions: The author indicated that “Our study shows a suboptimal rate of vaccine acceptance among Italian HCWs”. But the findings showed an overall vaccine uptake rate of 96.4%. How to define suboptimal rate of vaccine acceptance? We changed the sentence in the text.
Reviewer 4 Report
the authors have improved the manuscript but not satisfactorily. See few comments below
- Title: delete “(HCWs)” and “.”
- Lines 53-55: I disagree with the definitions provided by the authors. Currently, there is no such thing as “vaccine resistance”. The authors’ should access this recent article to see the correct definition of vaccine hesitance and then revise this part of the manuscript accordingly. Persisting Vaccine Hesitancy in Africa: The Whys, Global Public Health Consequences and Ways-Out - COVID-19 Vaccination Acceptance Rates as Case-in-Point. Vaccines 2022, 10, 1934. https://doi.org/10.3390/vaccines10111934
- Lines 107-111: The authors have listed the Italian regions surveyed but have not provided the method employed in selecting these regions in the materials and method section. Until the selection method is provided in the materials and method section, there is a bias; and this is not good for the study. If the regions were chosen purposively, then the authors should state the season for adopting purposive sampling method. The reason could be convenience, proximity to the authors, etc.
- During online survey, multiple entries are prevented by setting the internet protocol (IP) address to accept only one entry per device or email address. It’s important the authors know this for future studies.
Author Response
Thank you again for the you additional suggestions. We have improved the manuscript according to your comments.
- Title: delete “(HCWs)” and “.”
Done.
- Lines 53-55: I disagree with the definitions provided by the authors. Currently, there is no such thing as “vaccine resistance”. The authors’ should access this recent article to see the correct definition of vaccine hesitance and then revise this part of the manuscript accordingly. Persisting Vaccine Hesitancy in Africa: The Whys, Global Public Health Consequences and Ways-Out - COVID-19 Vaccination Acceptance Rates as Case-in-Point. Vaccines 2022, 10, 1934. https://doi.org/10.3390/vaccines10111934
We added the correct definition as requested and cited the upon mentioned article.
- Lines 107-111: The authors have listed the Italian regions surveyed but have not provided the method employed in selecting these regions in the materials and method section. Until the selection method is provided in the materials and method section, there is a bias; and this is not good for the study. If the regions were chosen purposively, then the authors should state the season for adopting purposive sampling method. The reason could be convenience, proximity to the authors, etc.
We added the information requested in the material and method section.
- During online survey, multiple entries are prevented by setting the internet protocol (IP) address to accept only one entry per device or email address. It’s important the authors know this for future studies.
We thank the author for the valuable suggestion, we will keep it in mind for future studies.
Round 3
Reviewer 3 Report
The authors have addressed the comments.